# The Comparison of Campus Planning Development at the Initial Stage of School Establishment: A Study of the Two Newly Instituted Private Universities of Science and Technology in Taiwan

**Chuan-Jen Sun [1,2,]* and Shang-Chia Chiou [1]**

[1]    Graduate School of Design, Doctoral Program, National Yunlin University of Science and Technology, Douliou, Yunlin 64002, Taiwan; chiousc@yuntech.edu.tw

[2]    Department of Visual Communication Design, TransWorld University, Douliou, Yunlin 64002, Taiwan

*     Correspondence: eric.sun@mail.twu.edu.tw; Tel.: +886-912-158-602

**Abstract:** Along with the adjustment of industrial structure, the pattern of the TVE in Taiwan has altered. Ever since the year of 1990 when the government started to promote new establishment of institute of technology and the institution status change, name change and upgrading policy of various types of schools, the number of technological universities and colleges has dramatically increased. However, in terms of a campus that affects essentially and greatly students' environmental cultivation learning, a technical college should in fact take the conformation of technical and vocational spirit and educational orientation into consideration and shape the school style and applicability of the technical university that possesses vocational education content based on school comprehensive planning and development framework of organic growth and sustainable operation. The purpose of this study is to probe into the course of campus planning development in the early stages of foundation of private university of science and technology in Taiwan. It is hoped that the research be conducted aiming at the development context of school formulation from the standpoint of technical and vocational education's historical development. This study adopts qualitative observation, documentary research and in-depth interview to try to understand the research topic from multiple aspects through field observation and interactive interview. Lastly, the study applies "comparative analysis approach" for reflecting on the development characteristics and issues of Taiwan's private vocational school in the light of campus developing process and implementation modality of private technical university before proposing the relevant suggestions at the end of the paper.

**Keywords:** campus planning; private university; private technical college; universities of science and technology; technological and vocational education; organic growth; sustainable operation

## 1. Introduction

### 1.1. Research Background and Motivation

Along with the adjustment of industrial structure, the pattern of the Technical and Vocational Education (or TVE for short) in Taiwan has altered. Ever since the year of 1990 when the government started to promote new establishment of institute of technology and the institution status change, name change and upgrading policy of various types of schools, the number of higher universities of technology has dramatically increased. The aggregate number has risen from 10 in 1996 to 78 in 2009, [1] (Table 1) which is equivalent to the increase of more than 7 times in just a few years.

Although the change and elevation of institution status of junior college is a compelling policy that needs executing for meeting the imperious demands of higher TVE's amplification in response to the social and industrial development at that time, the implementation ultimately causes the unanticipated consequence of disequilibrium between "quality" and "quantity" of higher TVE.

**Table 1.** Statistics of Universities of Technology & Colleges of Technology.

| School Year | University of Technology | | | College of Technology | | | Total |
|---|---|---|---|---|---|---|---|
| | Public | Private | Subtotal | Public | Private | Subtotal | |
| 1996 | - | - | - | 6 | 4 | 10 | 10 |
| 2001 | 6 | 6 | 12 | 11 | 44 | 55 | 67 |
| 2009 | 10 | 31 | 41 | 7 | 30 | 37 | 78 |
| 2017 | 14 | 49 | 63 | 1 | 10 | 11 | 74 |

However, in terms of a campus that affects essentially and greatly students' environmental cultivation learning, it is unlikely that a technical university can fit in with the pattern and scale of TVE by simply fulfilling or achieving the basic threshold of school ground area and total floor area of junior college reorganization regulated by Ministry of Education. On the contrary, a technical college should in fact take the conformation of technical and vocational spirit and educational orientation into consideration and shape the school style and applicability of the technical university that possesses vocational education content based on school comprehensive planning and development framework of organic growth and sustainable operation. Therefore, except for the majority of junior colleges which have transformed upon following the "restructuring model", those of a few public and private technical college that are founded by adopting "newly-established mode" are with significance and indicative meaning towards the approach of constructing a vocational campus that conforms to the "university" level at the time by comparison.

Until the school year 2017, the number of private colleges of technology (CT)/universities of science & technology (UST) has increased to 59; accounting for 80% of all the CT/UST in Taiwan and the number of students in those private CT/UST exceeds the total number of CT/UST students over 79% [1] (Figure 1). In this regard, the quality of the private CT/UST has a tremendous influence on the future workforce in our country. Consequently, with regard to a university campus, being both the representation of education and special carrier at the same time, the exploration of the concrete practice of school comprehensive planning and spatial organization performed by newly-established private technical colleges will help to reexamine further the developmental context and blind spots of Taiwan's higher TVE. This is exactly the major motive of this study.

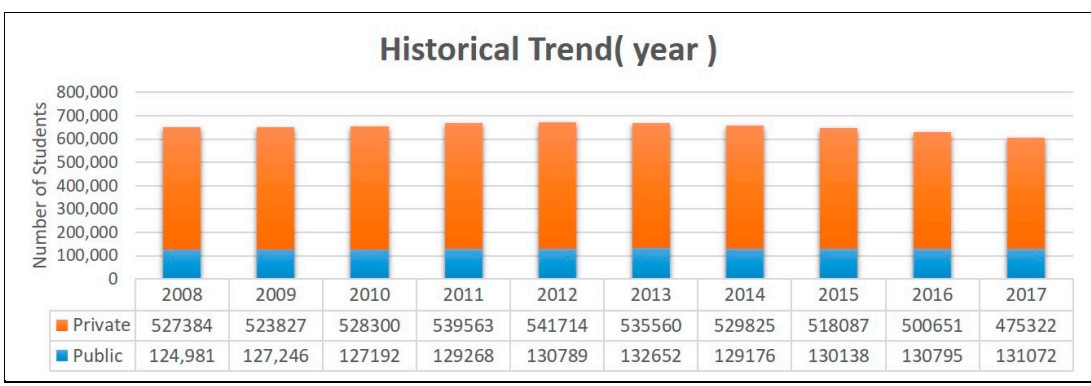

**Figure 1.** Statistics of Vocational Universities Enrolment over the years.

*1.2. Research Purpose and Method*

Based on the above background and motivation, the purpose of this study is to probe into the course of campus planning development in the early stages of foundation of private UST in Taiwan by

taking Chaoyang Institute of Technology (now: Chaoyang University of Technology or CYUT for short) (Figure 2) and Shu-Te Institute of Technology (now: Shu-Te University or STU for short) (Figure 3) as research subjects. It is hoped that the research be conducted aiming at the development context of school formulation from the standpoint of TVE's historical development.

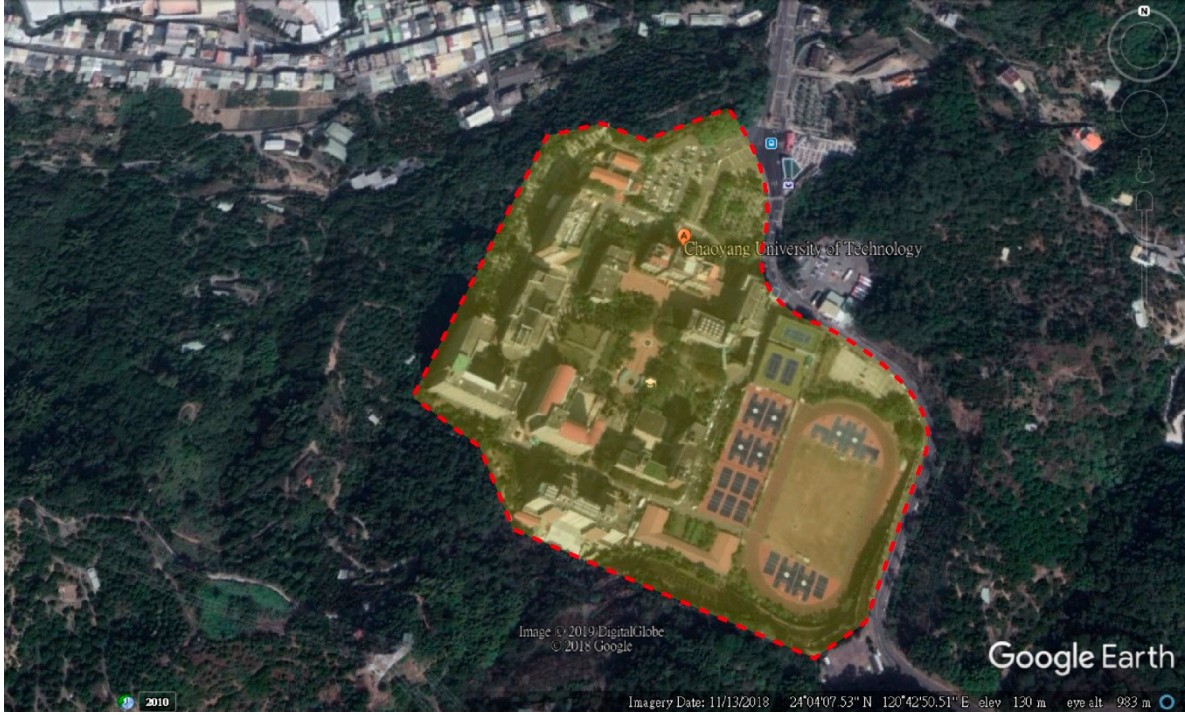

**Figure 2.** CYUT campus site [2].

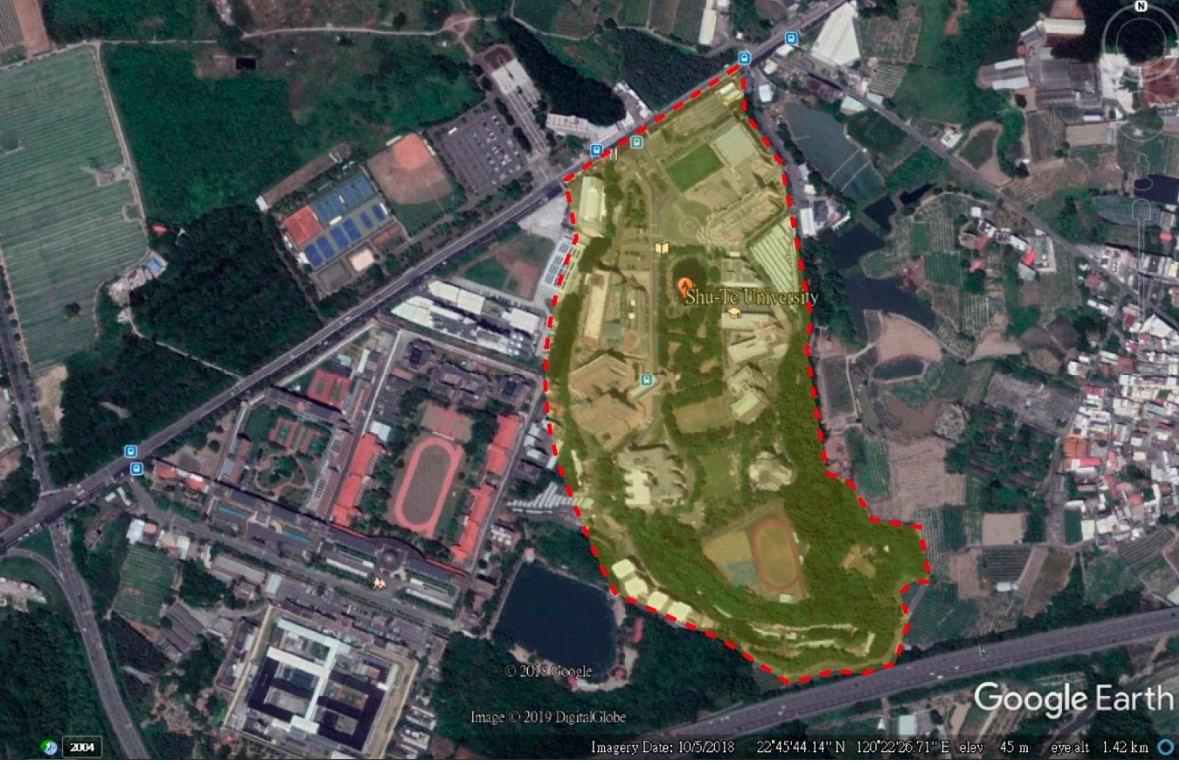

**Figure 3.** STU campus site [3].

This study adopts qualitative observation, documentary research and in-depth interview to try to understand the research topic from multiple aspects through field observation and interactive interview. For the background of interviewee, the interviewee is mainly the supervisor of the public works department of the preparatory department that undertakes and executes the business related to campus planning and hardware construction in the initial stage of the preparatory establishment. In addition, the method of "retrospective" is adopted to conduct in-depth interviews on the semi-structure of oral history.

Lastly, the study applies "comparative analysis approach" for reflecting on the development characteristics and issues of Taiwan's private vocational school in the light of campus developing process and implementation modality of private technical university before proposing the relevant suggestions at the end of the paper.

## 2. Literature Discussion

### 2.1. Development of the TVE in Taiwan and Private Education

On the strength of the idea of educational tracking, the system of Taiwan's higher education involves two categories of "university", the general education system and "technical school", the TVE system. In terms of the content of TVE, it mainly revolves around fostering all kinds of employment skills that the students are required to have at the workplace by combining the concept of Technological Education with Vocational Education. That is to allow students to "acquire knowledge, techniques and ability for job performance" [4] and to cultivate a large number of high-quality technical talents of various kinds and at diverse levels for the country by keeping up closely with the economic development strategies of our nations during different periods. This not only becomes the fundamental backbone of each industry but also constitutes the potential competitive advantages of Taiwan. Moreover, along with the changing times of economic environment and educational development, the developmental priority of TVE has varied radically.

Whereas the concern is about the development of before 1990, the government starts to deregulate, ease restrictions on higher education and shift the focus to the development of institutes and universities of technology that put emphasis on higher TVE after 1990. The whys and wherefores of the conversion is to respond to the appeal of university expansion sparked off by education reform when our government is confronted by the adjustment of industrial structure and the wave of democratization aroused by the termination of martial law. Meanwhile, in order to reduce the heavy education operating cost and financial burdens that come along with the institution of national universities (for example the investment of school buildings, faculty and resources of various types), the government vigorously encourages private education for boosting the investment of folk educational resources thereupon so that the huge expenditures in education can be shared. In order that the imperious demands of higher TVE's amplification of the society can be satisfied therein, the tactic of relying on the assistance offered by the establishment of private schools has inevitably become a necessary attempt that needs being put into practice. Since the most efficient way would be to select a great number of "exemplary" private junior colleges and transform them into institute of technology, the manifest development pattern of private schools' exceeding public schools in number and technical schools' being more than universities has become the development model of higher education in Taiwan.

To view from the background of educational development of Taiwan, as the stress of public schools' development lays greatly on the exam-oriented superior education, TVE which gives first place to the orientation of employment can only sink down to secondary education in the eyes of the general public. Hence the increase of private university not only realizes the dream of most Taiwanese children from middle-class or underprivileged families receiving high education, the private schools in Taiwan can also share the responsibility of universal education that needs to be fulfilled by public

schools essentially if we try to compare further with the condition of other countries. This shows the importance of "private education" to the education in Taiwan.

Taiwan's education reform has increased the number of higher vocational education institutions. However, compared to the overall higher education development environment in Taiwan, the higher vocational education is still in a less favorable situation. It is even more so for those that are private. There is a very unique phenomenon in Taiwan's higher education industry. The attributes of a college, such as public versus private or academic versus vocational, do not only affect teachers' perception and satisfaction with their working environment but also impact teachers' evaluations of their schools [5]. In other words, due to the fact that private colleges are generally not as competitive as their public counterparts in areas such as salary, benefits, rankings, resources and student recruitment, their teachers often feel being treated with less courtesy and respect which makes it difficult for them to have a higher evaluation of private colleges. Therefore, it creates an unusual phenomenon in which the flow of talents in Taiwan's higher education industry seems highly immobile. This does not only reflect the current situation of the improper allocation of higher education resources in Taiwan but also sheds light on the challenging situation faced by many private colleges. In terms of the financial subsidies for higher education in Taiwan, the ratio of government funding given to public and private colleges is four to six while the ratio of public to private college students is three to seven. Among them, private vocational colleges are the most discriminated against. The subsidy for private colleges has reached 20% but private vocational colleges received less than 15% [6]. This has seriously hindered the development of private vocational colleges.

On the other hand, with the rapid expansion of global higher education in recent years and the popularization of higher education, various impact and challenges have gradually emerged which affect the development of higher vocational education in Taiwan at three different levels: (1) global environment (globalization and cross-strait relations between Taiwan and Mainland China, (2) the social environment (the changes in industries and population structure) and (3) the educational environment (higher education system) [7]. In this research, they pointed out that "flexibility" and "sustainability" are two development trends that Taiwan's higher vocational education needs to pay attention to. Among them, the issue of "flexibility" mainly includes the flexibility of institutions and organizations [8], the flexibility of curriculum design [9], the flexibility of resource use [10] and the flexibility of learning outcome evaluation [11]. As for the issue of "sustainability", it does not only involve sustainable development of the world and the society but also that of the higher vocational education itself. It should be reflected in the development priorities of the higher vocational education during its adjustment period. In response to climate change, environmental and social changes, the subject of sustainability in higher education (SHE) has become an emerging research field particularly in recent years. The sustainable university has been defined by Velazquez et al. [12], (p. 812) as: "a higher educational institution, as a whole or as a part, that addresses, involves and promotes, on a regional or a global level, the minimization of negative environmental, economic, societal and health effects generated in the use of their resources in order to fulfil its functions of teaching, research, outreach and partnership and stewardship in ways to help society make the transition to sustainable life-styles".

Many studies [13–17] examined the increasing number of universities that had institutionalized sustainable practices in aspects such as courses, research, operations, outreach and assessment. Regarding the obstacles that affect higher education institutions' drive to promote sustainable development, Aleixo et al. [18] identified six important factors after reviewing many studies [19–28]:

(1)　the ambiguity and complexity of the actual sustainability concept, which is seen as an abstract and complex topic.
(2)　the lack of financial resources and funding.
(3)　resistance to change associated to behaviors, practices or initiatives.
(4)　the organizational rigidity of the structure (conservative, traditional and conventional).

(5)　　the lack of commitment, engagement, awareness, interest and involvement of faculty, students, staff, management and policymakers.

(6)　　lack of training and specialization in sustainability.

Aleixo et al. [18] also pointed out that the engagement of all the participants in the concept is the major driver. However, financial resources are clearly the biggest problem affecting the sustainable operation of higher education institutions. In particular, university presidents generally believe that in order to make schools sustainable, the biggest limitation is the lack of funds [29]. In addition, according to the results of in-depth interviews which Wright [30] conducted with the heads of the general affairs offices of 37 universities in Canada, it is also widely believed that financial austerity and sustainable management are the main issues facing universities in the past decade. Without sufficient funding, the goal of a green university will be difficult to achieve. If there is a new facility, it would require a separate fund for its operation and maintenance and "sustainable development" has been replaced by "money" as the university's top priority [31].

Although long-term and fixed funding is the most important issue for green universities, whether or not a university can operate sustainably is also determined by the number of students it is able to recruit [30]. This is one of the important issues that higher education institutions are currently experiencing in Taiwan. Due to the declining birthrate, the number of students enrolled in Taiwan's private vocational colleges has been declining in the past 10 years. In 2007, there were 660,771 students. By 2017, there were only 611,445 which is a decrease of 52,135 student enrollment in 10 years [32]. This shows that many universities in Taiwan, especially the private vocational schools, are bound to face more challenges caused by the declining enrollment rates. Therefore, a long-term vision for TVHE that takes into consideration of stakeholders who represent different aspects and attributes should be developed. The involvement of stakeholders is necessary to address their needs and long-term visions. [17]. In order to achieve sustainable development, attracting prospective students, making sure their educational philosophy is in line with the expectations of the society and the public, helping students find work after graduation and highlighting their distinct characteristics, have become the goals of many higher education institutions.

## *2.2. "Reorganization" of Junior Colleges versus "New Establishment" of Private Schools*

The 74 high technical and vocational universities are set up in succession between 1991 and 2014 by following either one of the two kinds of establishment modes: one is the newly instituted schools and the other is the "reorganization" of the existing junior colleges. In view of government's encouraging the original private "junior college" to transform gradually into "institute of technology" and head toward the educational policy of name change into "university of technology", it is not difficult to find that the 74 technical and vocational schools mentioned above are mostly founded by following the "restructuring model". Only five public and private universities of science and technology (National Yunlin University of Science and Technology, National Kaohsiung First University of Science and Technology, University of Technology, Shu-Te University and Yu Da University of Science and Technology) among are newly established. To compare the discrepancy between the two modes, it is obvious that the problems encountered during the preparation process of the newly instituted technical colleges are easier to work on due to the lack of former burden brought by the junior college, allowing the development of a more comprehensive scheme upon seeing the big picture.

On the other hand, with regard to technical colleges' change in name or status, the implication represented by a "university" also involves the accomplishment of the specific campus pattern that a university is required to possess over and above the transformation of substantive academic affairs including philosophy, curriculum, faculty and educational methods for high TVE. This is the framework that a university should present, the inclusion of concrete campus construction of all kinds of substantive content comprehending spatial function and scale plus environmental planning and design. Since a university is not just a place for learning and incubating high education for that the formulation of the campus is deeply influenced and reflected by the educational idea. Accordingly,

it is impossible for a vocational or junior college campus at ordinary level to draw a parallel not matter the size and extent of the campus or the complexity of spatial content. Therefore, how to satisfy the request of school buildings and teaching facilities or meet promotion standards of a university has become the priority that a technical college has to cope with when applying for reorganization as this is the development pattern a university ought to possess and the factor which determines the growth of the school in the long term.

The higher education policy in Taiwan is mainly based on the "University Act" and the basic entrance criteria for newly-established universities are implemented according to the "Standards for the Establishment of Universities and Their Branches" issued in 1996. In addition, privately-established universities also shall propose their own plans in accordance with the "Private School Law" and relevant regulations and shall raise sufficient funds by themselves and submit to the ministry of education for approval. In principle, the criteria for "school ground" and "school building" are the most direct impact on the overall planning of the campus. Therefore, the two standards are briefly described below.

The minimum school area standard for setting up a university shall be no less than 5 hectares for universities in the urban planning area and no less than 6 hectares for universities outside the urban planning area. This school standard is not only the smallest size a university should have but also an important key to the establishment of a university. Private universities are often constrained by limited funds and resources. The school area is basically in line with the minimum standards set by the Ministry of Education. Therefore, the area or location of the school is relatively unsatisfactory. To a certain extent, it will also affect the density and intensity of land use in campus planning. In terms of the space of school building, the floor space of the school building, which is required by the university, shall be set at a minimum of 20,000 square meters and shall meet this standard before the start of the new school year. In addition, the floor area of the school building must reach at least 12,000 square meters before the start of the first year of the school in the initial stage of the establishment of the approved project, to meet the needs of the school admissions. This requirement is very important for the implementation of the campus phased development and construction plan, the balance between financial stage planning and the actual development of school affairs and how to effectively construct a short-term, medium-term and long-term phased development plan to the overall campus planning.

*2.3. Campus Planning and Matters of University of Technology*

The planning pattern of a technical university campus falls into two types of "newly-constructed campus" and "reconstructed campus". "Newly-constructed campus" is the school established via preparations. Since the school ground is an undeveloped open space, the campus is required to complete the procedure of school institution through comprehensive planning. As for "reconstructed campus", the sort indicates generally the school with an existing campus but needs to map out a rehabilitation plan for expansion, reconstruction, renovation or enlargement of school ground owing to the transition or growth of school affairs. The two types mentioned above naturally differ from each other in purpose, function, content and operational approach.

For new universities, the campus planning should be based on the overall planning of the university's development needs, the use of the school ground and the construction of the school buildings. From the perspective of a new-founded university, as the thinking and consideration of "campus overall planning" is integrated during preparation, the "newly-constructed campus" is able to display a structure of consistency and integrity and define precise campus style with an environmental image which clearly delivers the vision and idea of education at the beginning of the establishment. By contrast, the private schools set up by following the "restructuring path" for junior college can only adapt to the actual situation of the original small school ground and carry out building extension or modification within the existing pattern because of having been unable to foresee the needs of development during the institution of junior college. On the other hand, due to the lack of campus comprehensive planning concept in early periods, the demands for school building often vary,

resulting in the space-filling chaos of campus in accordance with the development of school affairs. Apart from limited environmental resources and spatial development of school ground, the size and spatial function of the campus pattern can merely cater for the usage requirements of teaching of junior college.

Thus, in order to make up for the circumstances as mentioned above, private schools tend to adopt either one of the following two methods during the process of reorganization: one is to amplify the scope around the original school ground and the other is to look for new school grounds and build a second campus. Nonetheless, since most of the surrounding areas nearby these private schools have become prosperous along with the development of the school over the years, it is extremely difficult to broaden the range of the school ground in the vicinity. Consequently, the construction of a second campus seems to be more feasible that it has become the final decision made by many private schools.

There is no doubt that the heterogeneity of university campuses has been highlighted in many articles [33–35]. Due to the different land use and activities, the diversity and autonomy of various campus functions is similar to that of a small city. Therefore, the campus planning also involves designation and coordination of various zones that were traditionally adapted by urban planning and design. In fact, in addition to education, universities bear a profound societal responsibility [36]. This includes aspects such as training and educating the society, environmental management and the sustainable campus development [34]. Therefore, university campuses requires a holistic approach similar to that of towns [34]. In other words, the challenges a campus encountered such as transformation, expansion and management are similar to that of a real city which requires a specialized unit to establish a long-term comprehensive operational structure and procedure [37].

Especially since the "Stockholm Declaration" was put forward in 1972, people have deeply recognized the human interdependence with the environment. Sustainable development has gradually become the common goal of all countries. Therefore, many universities started to bring in the concept of sustainability to their campuses [38], Taiwan is no exception. Many universities around the world have begun to promote the concept of "green campus" to facilitate the sustainable development of the campus. The green building initiative represents a sustainable design concept which consists of a series of projects designed to minimize the production of waste and hazardous materials, reduce level of energy consumption and promote the design of energy-efficient buildings [34]. Since 1996, in response to the global trend of sustainable development, the Taiwanese government announced the comprehensive promotion of green building policy in the "Construction Policy White Paper" and launched the "Taiwan Green Building EEWH Assessment System". In 2002, the "Green Building Policy" was included in the "Challenge 2008 Six-Year National Development Plan" to implement Taiwan's sustainable environmental policy. The EEWH assessment system focuses on ecology, energy saving, waste reduction and health. It is currently widely used in Taiwan for green building assessments. Overall, effective sustainability indicators play a corresponding role in the sustainable development of campus planning, especially when considering items such as campus land use, landscape greening and open space retention and campus building configuration and form in the early stages of campus planning.

In addition to the construction of physical space on campus, sustainable education is also an important part of sustainable campus development. In particular, after Taiwan experienced the "921 Earthquake" in 1999, the Ministry of Education of Taiwan (MOE) carried out large-scale campus reconstruction and new construction and began to rethink the issue of sustainable development of the campus. Since 2000, the Ministry of Education of Taiwan (MOE) has started to promote the "New Campus Movement" and the "Taiwan Sustainable Campus Program (TSCP)". Furthermore, by combining the perseverance education with the environmental education, the MOE hopes to instill the concept of sustainable development and environmental protection in the younger generation from an early age. In other words, it is an innovative approach which uses the environmental education as the basis of thought and green schools as a means of design. By combing the two, the concept of environmental protection will be sustainable and integrated into the education system [39]. It can be

found that on the basis of the sustainable development of the campus, the humanistic concept [40], the sharing of resources and the connection of local neighborhoods [39] provide a deeper reflection on campus planning. As Alshuwaikhat and Abubakar [34] pointed out, "universities should promote a pattern of development that would be compatible with a safe environment, biodiversity, ecological balance and intergenerational equity".

### 2.4. University Campus Planning and Design Theory

The concept of the overall campus planning concept was not really taken seriously until the end of the 1940s after World War II. Among them, Richard P. Dober's four series of campus planning series have systematically and comprehensively discussed three aspects of planning, architecture and landscape and successively proposed "Planning Modules" and "Building Standardization System". Concepts such as "campus image structure" have become an important reference for many campus planners. In particular, Dober [41] in the book "Campus Design" discusses the main planning and design elements that constitute the image of the university campus and puts forward two viewpoints such as "Place-making" and "Place-marking". It considers that "land planning and use", "traffic network system", "building location configuration", "campus physical environment" and "campus infrastructure facilities" are the basic requirements for the "place creation" of campus hardware. The "landmarks", "landscapes", "materials", "styles" and so forth are the basic elements of the "software-level" campus software to perform "Place-marking". It believes that it is necessary to try to construct and shape the image of the campus and give it the meaning of the place to create a high-quality campus environment.

After the 1970s, the campus planning, due to the influence of urban design and other related reflections, began to abandon the one-time overall design and the pursuit of static external forms and paradigm shift direct planning and design evolved into a "dynamic progressive planning" that respects and emphasizes indirect norms. As Christopher Alexander, et al. [42] in his book "The Oregon Experiment", through the urban design concept, it is directly applied to the interpretation of campus planning in a simulated empirical way; it believes that the most important aspect of campus planning is the integrity of the process and the six important ideas put forward have not only become an important theory of modern campus planning but also the three concepts of "Organic Order", "Piecemeal Growth" and "Patterns" have made the campus master plan more reasonable. The basis of the design. In addition, Kevin Lynch [43], in his book The Image of the City, proposes five basic elements that make up urban imagery: paths, boundaries, regions, nodes and landmarks. These elements together constitute the spatial structure of a city. It is not only an important reference for urban residents as a perceptual environment but also a basic design principle for urban design. It can also be used as an important reference model for examining the shape of campus space.

In recent years, some researches on the shape of university campuses can be found that most of the conceptual foundations are based on the research results of the more mature urban forms and try to explore the general campus morphology research from the two aspects of social form and material environment [44–48]. In theory, the general university campus environment includes the physical environment and the spiritual environment of the campus; and this is also a two inseparable component of a good campus. The former includes the school sites, school buildings, campus, playgrounds and ancillary facilities. It is the spatial entity environment of the elements of the school covered by material culture; the latter refers to the core idea of the university spirit, that is, the vision of the school, the idea of running a school and the educational concept. The humanistic environment formed by the comprehensive human factors. The two complement each other, are indispensable and interact and restrict each other and together constitute the so-called university culture. As Pearce [49], (p. 32) thinks: "Education itself is intangible and architecture gives its material form." In other words, each university has its own unique culture, which is the sum of its material and spiritual achievements. It is accumulated in the historical evolution of university development, imperceptibly permeating and

influencing every level of the university and inheriting the foundation here. In the past, through the performance of the campus physical environment to emit a distinctive temperament.

With the concept of sustainable development being applied in universities, a clear vision and management's commitment to sustainable development have become even more important [34]. In other words, for new universities, the vision and the planning framework of school development is the consensus among the stakeholders [34,50]. It is used to determine and outline the future development of the campus. Furthermore, it provides a direction for campus planning and school building design and drafts a blueprint for the future campus.

In recent decades, many studies have discussed the involvement of different stakeholders in the campus planning process which adds new qualities, both physical and spiritual, to the overall vision of the campus. Efforts include the following initiatives: Through the process of public participation, campus stakeholders and the local community work together to gain a deeper understanding of the needs of stakeholders' and commit to develop a model of sustainable campus which is combined with the University Campus Master Plan to effectively achieve the goal of sustainable campus development [50]. Through participatory design, stakeholders are involved in the design concept of EcoHouse and can better understand the real needs of campus users and their views on sustainable campus life [51].

There is a common understanding in the literature, that is, stakeholders' participation in campus planning is indeed helpful to understand and meet their actual needs and is effective and positive for promoting sustainable campus development. A sustainable university campus implies a better balance between economic, social and environmental goals in policy formulation as well as a long-term perspective about the consequences of today's campus activities [52]. Therefore, "safeguarding environment", "economic management" and "social justice" [34] are three factors which should be taken into serious considerations when formulating university policies, visions and goals. However, the case of public participation in campus planning in Taiwan can be found mostly in public universities and rarely in private universities mainly due to the technical operation and implementation level. A lot of complexity and uncertainty have caused private universities to avoid adopting such method during campus development.

The campus model defined by the five planning principles mentioned in a study done by Ribaraygua Batalla and Garcia Sanchez [51] provides a great reference for campus planning and design process for promoting sustainable campus development. The five planning principles are as follows:

1. Social Learning Campus: Creation of a sustainable network of open spaces
2. Integral Campus: Combination of new university uses
3. Accessible Campus: Campus of Sustainable Mobility
4. Didactic Environmental Campus: Biodiversity and consolidation of botanic routes
5. Campus Morphology: Creation of a campus landscape into its surroundings

Based on the above, campus planning is a design method and means for pursuing and practicing the "good university campus form" and attempts to shape the ideal appearance of the campus form with macro concepts. From the perspective of the development of the campus, the campus is an organic form and its most fundamental feature lies in the "wholeness" that the essence presents. The campus is more like a miniature of a city. Regardless of the development of the spatial structure and the planning method, the urban design and campus planning have been used for reference from time to time. Huang [37] believes that university campus planning is a dynamic planning process and should be considered as an urban design paradigm. Therefore, for the newly established university, in order to lay the foundation for the continuous growth of the university campus and to provide the foundation for a benign development of the university, the campus planning must be considered from the overall level of the campus structure, structure, function, modeling, style and so on. And to seek continuity in time and space. In other words, in a certain period of development, the overall campus-related land

use, spatial layout and various facilities are planned and managed in an integrated manner, aiming at setting and formulating guidelines for the goals of campus development and construction.

Therefore, under the long-term control of the university's overall development goals and the development of the university's development orientation, the main projects of the "design stage" of the campus's overall planning and design structure can be summarized into the land use, site plan, road system, campus landscape and open space. Based on the above, the subsequent case analysis and comparison of this study will be discussed in six aspects: Concept of Foundation and Development Orientation, Middle and Long Range School Development, Land Use, Site Plan, Road System, Open Space and Landscape.

## 3. Case Analysis and Results

### 3.1. Case 1—Chaoyang Institute of Technology

#### 3.1.1. Concept of Foundation and Development Orientation

The pioneering days of Chaoyang Institute of Technology's preparation fell upon the year of 1985 when the Ministry of Education reopened the institution of private school. Established and begun its enrollment in 1994, Chaoyang Institute of Technology becomes the first new-founded private technical institute not converted from a junior college in Taiwan. Being the only private school among the first five technical schools that rename "University of Technology", it is not only the representative of vocational college but also the first case that adopts "newly-established mode" of private school. The concept of foundation is "to achieve the objective of graduation means employment and get into the swing of things at work" based on "cultivating top-quality talents that own expertise, professional skills and career ethics". What makes the school special is that it applies the establishment of only five departments, which are department of construction engineering, department of information management, department of architecture and design, department of industrial design and department of industrial engineering and management, during the period of preparation. We can discover from documentary data analysis and interview that the aforementioned items apparently correspond to the job attribute and personnel requirement under Ever Fortune Industrial Co., Ltd., which reinvests to set up the school and accordingly become the enterprise-based school-running mode rarely seen in Taiwan. The business itself also authentically engages in the project of campus formulation and construction that it not only guarantees and provides internship and employment opportunities but also indirectly brings up professional and technical talents for school-running enterprise. In a way, this is closer to the pragmatic intention of TVE instead.

#### 3.1.2. Middle and Long Range School Development

On account of the purpose of establishment, which is to articulate and continue TVE, instead of partitioning and setting up faculties or colleges during the initial period, the scheme of undergraduate two-year and four-year college school system is adopted, together with the development of graduate institute starting from the second year. The objective is to develop into a comprehensive institute of technology with the campus scale of 6000 to 8000 students (see Table 2), based on which the comprehensive planning and design for campus is carried out.

**Table 2.** Statistics on the number of students in CYUT from 1994 to 2003.

| School System \ Year | 1994 | 1995 | 1996 | 1997 | 1998 | 1999 | 2000 | 2001 | 2002 | 2003 |
|---|---|---|---|---|---|---|---|---|---|---|
| 4 year programs | 663 | 1387 | 2227 | 3185 | 3697 | 4353 | 5201 | 5813 | 6487 | 7026 |
| 2 year programs | 273 | 594 | 688 | 909 | 1121 | 1195 | 1322 | 1314 | 1083 | 1109 |
| Master's program | - | 18 | 82 | 140 | 200 | 295 | 408 | 527 | 691 | 827 |
| Doctoral program | - | - | - | - | - | - | - | - | 3 | 6 |
| total | 936 | 1999 | 2997 | 4234 | 5018 | 5843 | 6931 | 7654 | 8261 | 8968 |

### 3.1.3. Land Use District

Located in Wufeng District, Taichung City covering a measure area of 27.38 hectares, Chaoyang Institute of Technology is a semi-open vantage point that stands on the hillside. Limited by the hillside terrain, the practical usable area of the campus is finite and short of the possibility of expansion. Thus, in regard to land utilization, the maximum economic benefit of the overall development of the school is taken into account that four use districts are roughly divided according to diverse functions and attributes (Figure 4). In practice, the school tries enhancing the development strength of the land to enable the maximization of beneficial result of school development in order to guarantee and keep the completeness and presence of an appropriate amount of public open space and green space on the campus. As for teaching and administrative districts, the buildings are centralized, together with intensified volume (high rising buildings) planned.

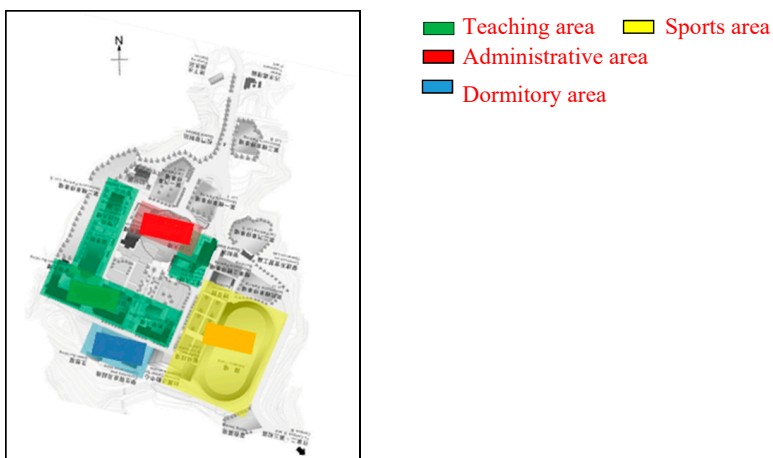

**Figure 4.** CYUT campus land use division.

### 3.1.4. Site Plan

The development axis of the campus depends on the southwest facing northeast slope terrain after school grading and the layout of the clustered buildings corresponds horizontally or vertically to this axial growth (Figure 5). The buildings configured separately on both sides of the path center on the main circular road, forming a linear spatial pattern by the encompassment of several clustered buildings within the administrative and teaching districts. Meanwhile, the spatial structure of "line cluster mode" that allows the shaping of a legible and plain fabric of the entire campus is revealed by the imaginary axis of the central landscape trail formed by the combination of administration building stretching to student dormitories. The administration building, which lies at the terminal view of the axis, naturally becomes the high self-evident landmark of the school. As to sports and recreation facilities, they are disposed on the east side of the campus, separated from teaching district by the circular road to attain dynamic-static segregation of space attribute.

### 3.1.5. Campus Road System

Since the hinterland of the campus is rather small and narrow and for avoiding while reducing the interference brought by the traffic flow to the central area of the campus, the formulation concept applies administrative district as the boundary line to seclude visiting vehicles on the periphery of the center zone. In addition, the school also spares no effort to maintain and make the central area to become a local "traffic calming zone" that will never be disturbed by the running vehicles on campus (Figure 6). The center zone of the campus itself adopts the idea of pedestrian and vehicle separation that the road system falls into three main categories. Besides siting sidewalks on both sides of the "circular road", the blocks it surrounds form the "regional type" pedestrian campus where only walking is permissible. Apart from ensuring the safety of teachers and students who tread around the

campus while avoiding the disturbance of traffic flow, the system links up each open space with all activity nodes to offer people activity venues, recreation and gathering spots for creating opportunities of manifold participations and interactions.

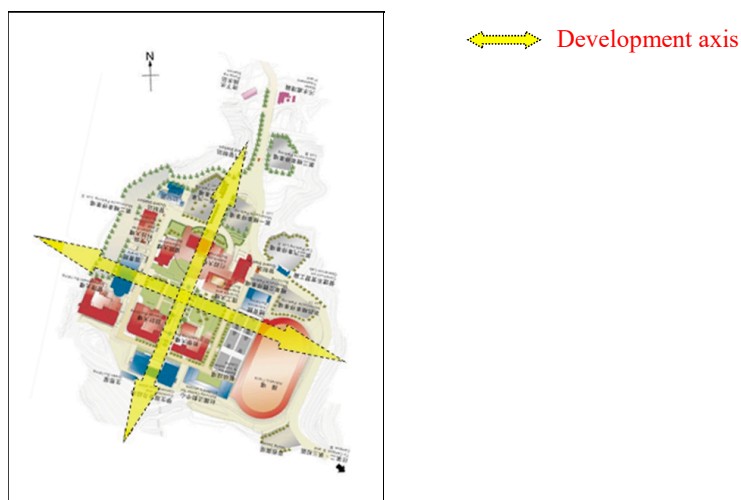

**Figure 5.** CYUT campus development axis.

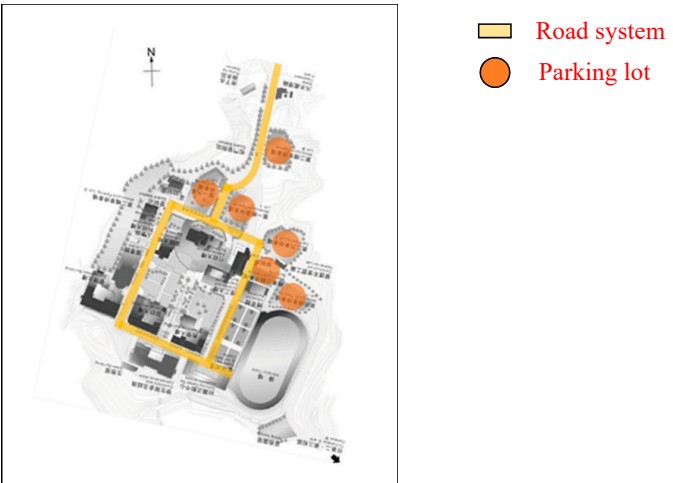

**Figure 6.** CYUT campus road system and parking.

3.1.6. Shaping of Campus Landscape and Open Space

For external space planning, central landscape axis proceeds to organize, connect and vary successively and sequentially by means of the virtual and real spaces on school grounds, enabling a particular spatial sequence of the overall environmental landscape of the school that can assist teachers and students in building the image of campus space. What is more, the hillside terrain is able to add interest to the change of space volume. Nevertheless, as most of the clustered buildings assume centralized and high-level design, especially those with mega mass along the sides of the circular road, the specific value of D/H is evidently lower than the desired value. As a result, a sense of oppression produced by the spatial scale consequentially follows. Furthermore, owing to its mountain locality, the school has only one primary entrance joining the access road that leads to the city under the mountain on the north side. Since this is the only connection to the outside world, the liaison and fusion of the school with town and urban areas is relatively poor.

### 3.2. *Case 2—Shu-Te Institute of Technology*

#### 3.2.1. Concept of Foundation and Development Orientation

Established and begun its enrollment in 1997, Shu-Te Institute of Technology is the first new-founded private technical institute not converted from a junior college in southern Taiwan and the second private technical college that is set up by adopting "newly-established mode". The concept of foundation is "to implement the pragmatic spirit of fostering talents for our country, improve the quality of TVE system and promote the upgrading of commerce, service industry and design service industry". This individual case, in the main, is the University of Technology established separately by the board of directors of private Shu-Te Home-Economics and Commercial High School. The purpose of its institution is to follow the idea of all-around education upheld by the original the board of directors of Shu-Te Home-Economics and Commercial High School. More precisely, the vision is to increase learning capability and employment of technical personnel and to develop further into a comprehensive university with the goal of articulating and continuing TVE through diversified, integrated business model reposed on southern Taiwan.

#### 3.2.2. Middle and Long Range School Development

Founded in response to the permission of Ministry of Education's easing restrictions on the application of vocational institutes with school grounds of more than 10 hectares in 1994, the project schemes out its school system on the basis of undergraduate two-year and four-year college in principle during the early stages. The development of graduate institute then takes place after the constructing completion of undergraduate section. The newly established five faculties of four-year undergraduate program in the first year include "Department of Information Management", "Department of Business Administration", "Department of Interior Design", "Department of Leisure Management" and "Department of Applied Foreign Languages", together with another three faculties established in the second year. Aside from initiating three additional faculties for the four-year college, a new faculty for the two-year college is also set up starting from the third year. The construction of the integral undergraduate system, which involves 11 departments in totality, completes in the fifth year followed by the commencement of siting two graduate institutes. The objective is to turn into a medium-sized university campus in middle and long term and to reach the estimated scale of 6000 students in the long run in accord with the school development drawn up at preparation stage, based on which the overall planning of the campus for this case is followed out.

#### 3.2.3. Land Use District

Located in Yanchao District, Kaohsiung City covering a measure area of 16.64 hectares, Shu-Te Institute of Technology has been troubled with the problem of usable area's being short of possibility of expansion over the years. Due to the limitations caused by earthwork leveling and soil conservation, which lasted for more than five years, during pre-exploited period plus the highest hill at an elevation of 172 meters, it is crucial to utilize and control the land properly and efficiently. In view of development benefit and operation cost, the campus is subdivided into four use districts of dissimilar features and attributes including administrative and sports division, teaching division, dormitory and residential division and landscape lawn area according to diverse functions, properties and the aspect and grade of slope terrain (Figures 7 and 8). In principle, the formulation adopts approaches that are more economical as far as possible. In addition to reducing development intensity for keeping the completeness and openness of the green space on the hillside, a large-scale linear landscape belt is set to serve as a complete open space at the center of the campus and to partition the extent of teaching division and dormitory on both sides. The administrative area, which tends to have more contact with the outside world and sports area, which is open and with high usability, are built on the street frontage to act as the cushion space that insulates noise from Cinan Road to enable the formation of a dynamic-static space pattern of the campus. Buildings in other zonings are centralized, together with

the construction of intensified building mass (high rising buildings) formulated to produce bigger floor area.

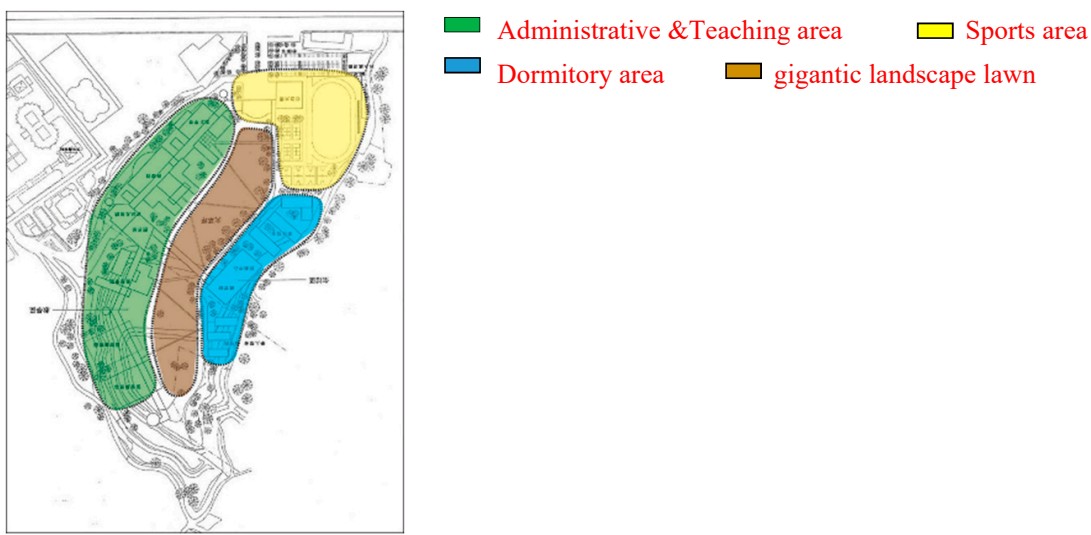

**Figure 7.** STU campus land use division (original plan).

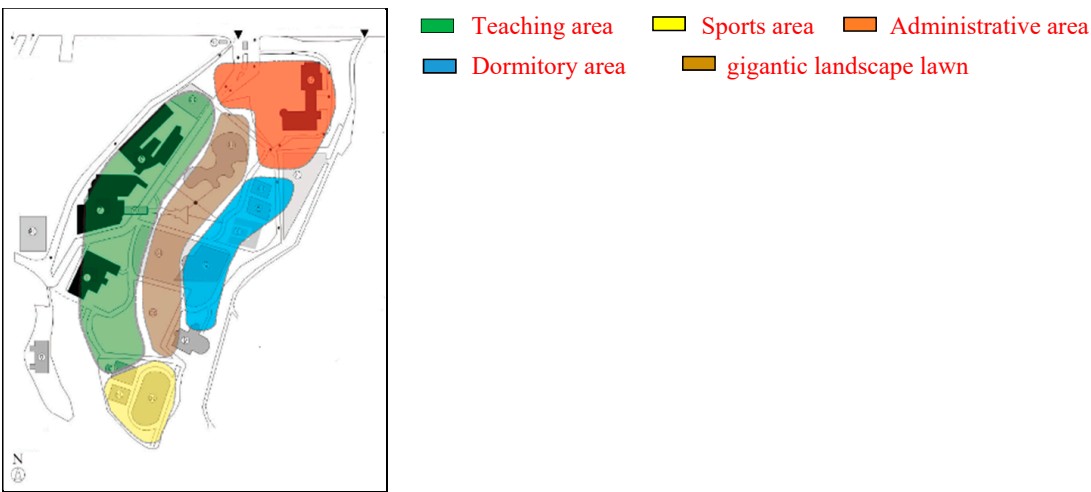

**Figure 8.** STU campus land use division (adjustment plan).

3.2.4. Site Plan

The campus is a site in a narrow and long, irregular and winding shape and the development axis of the campus, which forms the southwest facing northeast slope terrain after school grading, deduces a spatial structure of "linear banding mode". Whereas the teaching division centers around the circumferential road on the west side of the campus and stretches out along the linear road where several centralized buildings tightly adhere, the residential area centers on the secondary pedestrian walkway on the east side of the campus. The whole of the campus is the cohesion of buildings on both sides sphering the large external open landscape in the middle, resembling the Yard System of Academic Village often seen in an American campus space in the early years. According to formulation layout, the teaching area reveals a banded space of organic growth that extends along the circumferential road on the west side of the campus. Within the range allocating the buildings of each faculty and school based on centralized and high-level design and the ties among are established gallery axis. Spaces like gymnasium, student activities center and dormitory, whose attributes are highly related, are concentrated to form dormitory and residential area and configured on the east side of the campus for avoiding interference with the teaching division. The communities of the school

buildings are in the main isolated on the two sides of the large lawn in the center of the campus to ensure that every school building is well ventilate with natural daylighting and excellent view. (Figure 9).

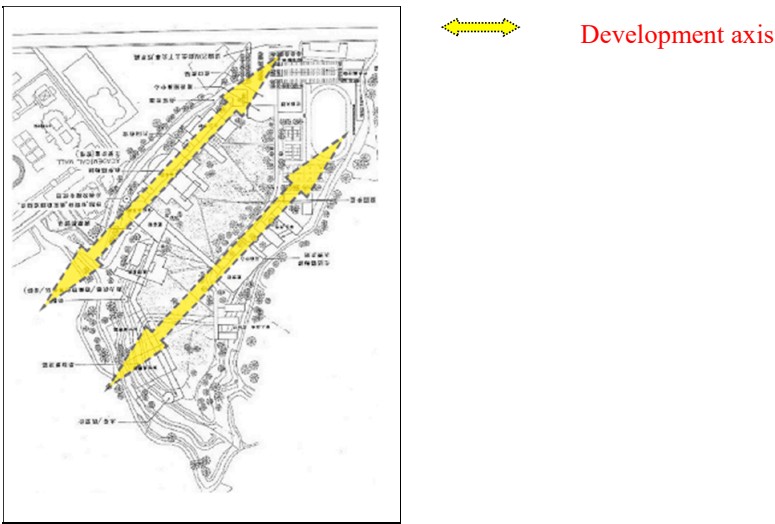

**Figure 9.** STU campus development axis.

### 3.2.5. Campus Road System

This project adopts the formulation idea of pedestrian and vehicle separation and by following the concept of "mainly rely on people supplemented by cars", the road system falls into two categories (Figure 10). The first is the circumferential road on the west side of the campus, which is the principal traffic route that links up administrative division and teaching division. The pass of the buildings within the teaching area connect one another by a semi-outdoor skylight corridor that leads all the way to library and dormitory area. This corridor, which is taken to be the pedestrian-based "trail system", is the second road system built for guaranteeing the safety of teachers and students who tread around the campus while increasing communications and interactions. On the linear landscape belt in the midst sited many irregular jagged walking paths of 1 to 3 meters wide to join dormitory area and teaching area. Besides keeping basic transportation function, the purpose is to reduce, shorten the traffic flow of automobile on campus and seclude the running vehicles on the periphery of the campus. The school also spares no effort in maintaining and making most part of the campus to become a local "traffic calming zone" that will never be disturbed by the running vehicles in order to build up a "walking campus" exclusively for pedestrians in the concept of planning. However, due to the cancellation of semi-outdoor skylight corridor and central lawn trail after the institution, the idea of pedestrian-oriented "integral type" walking campus as originally schemed is rather impracticable under realistic conditions.

### 3.2.6. Shaping of Campus Landscape and Open Space

As the exploitation of campus' school grounds takes place after massive earthwork excavation, backfilling and land leveling, some soil locations of the base are not appropriate to dispose buildings for reason of security concerns. For example, the gigantic landscape lawn, whose size is 600 meters long and 100 meters wide in the midst of the campus, actually situates at the junction of two types of soil. As a result, the formulation deliberately avoids the area and accordingly molds the mega banding lawn area, which not only act as an open space and ecological green landscape zone but also the sustainable availability of an adequate, pleasant scale space, in the center of the campus. As regards the other clustered buildings, they alter along the axial geography by following the slope grade layer upon layer, displaying a spatial sequence of continuity order ability. Nonetheless, since the campus is a

self-contained area which has only one primary entrance with the width of 80 meters connecting to the outside world on the north side plus its suburban locality, the liaison and fusion of the school with city and urban streets is relatively poor. In the meantime, the sports and activities area adjacent to the main entrance in the north of the campus as originally planned moves to the southernmost sloping field of the school at the initial stage of school establishment according to plan modification. Hence, because of school's remote geographical position, the accessibility and convenience of sports venue and facilities that can be open to and shared substantially with community residents are rather weak.

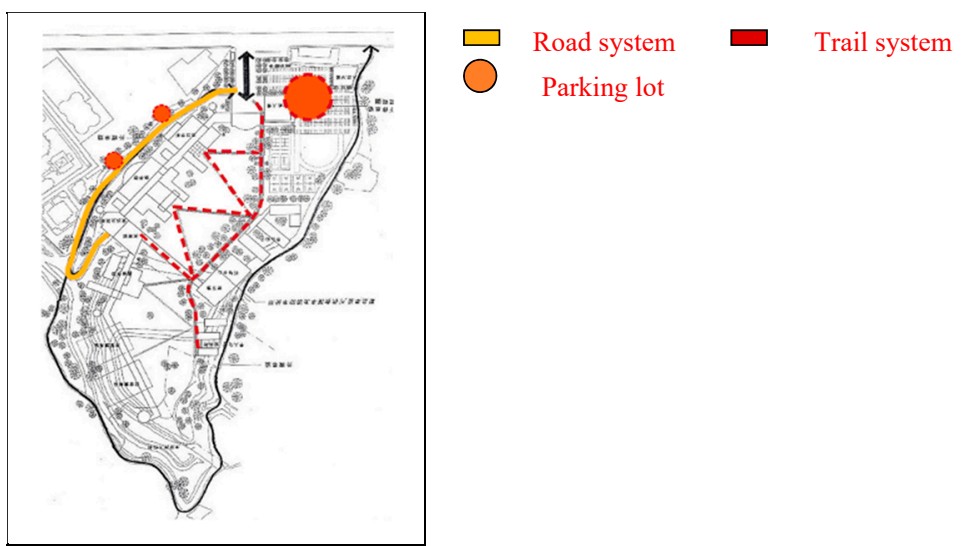

**Figure 10.** STU campus road system and parking.

## 4. Comparative Analysis and Discussion

Through comparative and inductive method, the findings are as follows:

### 4.1. The Clear School Orientation

In sight of school's development vision, which is to determine and draw the outline of what a university campus is going to display and develop in the future, not only is the distinctive school style created but also the appearance of the campus which is going to be converted into a university in days to come constructed. It is imperative that the establishment of "school vision" return to the most fundamental positioning thinking and mission definition of the school, the original starting points of school institution as well as the formation of the character of university governance of the newly established school. According to research findings, it is obvious that instead of having a substantial or modal mission and orientation like public universities, private universities tend to be more "educational market-oriented". Consequently, the intention of school founders or the board of directors and the trend of educational market are prone to be the principal driving factors. Moreover, we also discover, from research interview, that the dominant influence of the director of provisional office varies because of being subject to the inclinations of the supervisors (founders or the board of directors). If the director can acquire full authorization, the execution attitude of campus comprehensive development and planning will be able to reflect the educational idea. Mentioned in the interview:

> *President Zeng has his own philosophy on running a school. I think there is no one else who knows more about this school than he does. Therefore, many of his ideas and visions for the future will certainly be incorporated into our school planning. (CYUT Visit-A-1:23)*

*In my opinion, the contributions made by President Zeng is one of the main reasons why Chaoyang is doing so well today. (CYUT Visit-A-1:24)*

*4.2. The Campus Scale with Planned Dynamic Growth*

We can learn from literature and interview data that for most of the junior colleges or new investors of educational business, they are inclined to adopt a "wait-and-see" attitude as higher TVE pattern is still a brand-new concept or educational policy with many strict application requirements and procedures to grope for at that time. The tendency of conservative and prudent approaches are also found with regard to school size and middle and long-term school development that they are most likely to refer to the draft and formulation of a national university which has been founded earlier, for example National Yunlin University of Science & Technology. The development objective for school size and middle and long-term school development is to achieve a scale of 6000 to 8000 students in an attempt to become a medium-sized comprehensive institute of technology. Mentioned in the interview:

*For private colleges, it is inevitable for management to use a business mindset to run a school. Of course, the market determines what programs we offer. We need to make sure that there are students who would pay to attend our school so that we don't lose money. (CYUT Interview-A-1:11)*

*Overall, I hope that this school can develop into a medium-sized school. It will mainly focus on recruiting students for the four-year and two-year programs in the beginning and will start recruiting graduate students in the fifth year. (STU Visit-B-1:28)*

*In the past few years, our school has done a good job and has grown very fast. We have a doctoral-level class, 12 master-level classes and 18 departments. The number of students is about 12,000.*
*(STU Visit-B-1:41)*

*4.3. The Campus Space Resources for Rational Distribution and Effective Use*

The development and utilization of school grounds must take into account the economic benefits of development and the sustainable development of the environment, in order to create the campus space resources for rational distribution and effective use. As most of the new-founded private universities repose on the hillsides in suburbs conforming to or exceeding the standards of basic threshold enacted by the Ministry of Education, the practical usable area of the school grounds, often limited by grade, are finite and short of possibility of expansion in spite of covering an area of 15 to 30 hectares. Thus, in regard to land utilization, school buildings tends to be centralized, together with the construction of intensified building mass (high rising buildings) formulated to produce bigger floor area and to keep the completeness and openness of an appropriate amount of public open space and green space on the campus. Mentioned in the interview:

*...., the land is actually limited, how to make good use of the limited land is very important, ... we roughly divide the school into several zones so that the teaching buildings are concentrated in this area and the large open space or sports field is placed on the other side and the rest will probably be used for parking and some landscapes. (CYUT Visit-A-1:56)*

*When we are doing it, we use it according to the partition. It is like our road system, the public power and the drainage of the campus. Some facilities have likely already been reserved in advance... (STU B-2:37)*

*4.4. The Integration of Functional Layout of Teaching Areas and Intensification of Spatial Structure*

For the planning structure of spatial model, the benchmark of developmental axis is set up complying with the slope terrain of the campus, which combines functional spaces of the same attribute into various spatial clusters that are concentrated and configured on both sides of the linear reference road. Two types of spatial structures, the "linear banding mode" and "linear cluster mode", conclude

a part of the layout. Besides, due to relatively cramped usable areas on the hillside, the enhancement of land use intensity is the strategy assumed by many private schools. Hence, in principle of claiming larger floor area while reducing floor space of the school buildings, both universities are inclined to discipline composite and high-level volume design. Although both cases do not have explicit school or faculty conceptual compilation, the disposition of school buildings in teaching division, through centralizing spaces with teaching function and integrating the disciplines of the same attribute inside a mega building complex, is arranged on the basis of a vertically compound "partial centralized configuration". By means of the management and design of function and facility intensification of the internal space, the effectiveness of land resources use on campus is improved. Meanwhile, thanks to the effective management of teaching facilities, the sharing of teaching resources and lateral exchange among diverse fields are empowered. Mentioned in the interview:

> *... mainly because the school is small, it has to be raised upwards in terms of volume, ..., of course, the configuration is more concentrated but because this is a hillside, the front and rear rows will have a height difference and the views have not been blocked. (CYUT Visit-A-1:58)*

> *Every building is now a college. Because the functional properties are similar, they are all put together. They are mostly facing either south or north. (CYUT Visit-A-1:59)*

> *..., because the school itself is not large and its topography and geology is quite unique, it requires the use of larger bodies to deal with them. Therefore, in the building configuration, the buildings are mainly along the road and leave the large piece of green open space in the center. (STU Visit-B-2: 52)*

*4.5. The Efficient Campus Road System, Humanization and Diversification of Communication Space*

The rational and user-friendly overall formulation of "campus road system" guarantees not only the accessibility and convenience of floating population and traffic flow but also constitutes the spatial framework and integrative order of the campus through organic connection between school buildings and each land district based on facilitated road network. In principle, the formulation applies approaches that are able to maintain a user-friendly environment on campus and live up to the traffic pattern of "mainly rely on people supplemented by cars" as much as possible in accordance with the planning idea and conception of "taking human beings as the essential". Meanwhile, the school chooses road network type suitable to local conditions by cooperating with land use district and building layout and adopts a road system with construction schemes of distinct levels. The purpose is to make sure that a "traffic calming zone" that will never be disturbed by the running vehicles is marked off in the central area of the campus in order to build up a "walking campus" exclusively for pedestrians. In the case of school parking system, both cases act according to the principle of "the combination of centralization and decentralization" and carry out apparent planning strategy of "campus peripheral development" in practice. The main purpose is to have all teachers, students and visitors park their vehicles in the parking lot outside the campus before entering the campus. This is to ensure the safety of the pedestrians and avoid campus events being disturbed by the passing vehicles. Combining open spaces and providing a resting and gathering place for people creates more opportunities for people to interact with one another. In other words, it is to avert excessive concentration setting and disperse evenly in the surroundings of the road in every district, to deploy in the surroundings adjacent to the entrance or exit of the campus and roads on the periphery of the campus for encouraging walking to the central area of the school after parking. The ultimate purpose is to retain the safety and tranquility of the center zone and relief load pressure born by the road system and parking spaces on campus. Mentioned in the interview:

> *Because the area itself is not large and it is on a hillside ..., the roads inside the school have become very important, ..., make it a two-way inner ring road, just around the hillside for the vehicles which is the main moving line of the campus. In fact, it is very simple and clear and won't waste too much space to build the road. I think it is very economical. (CYUT Visit-A-1:42)*

*In the inner ring road, there are lanes for people and vehicles and there are trails along both sides of the road. This pedestrian trail can just be connected to the middle area I just mentioned. It is generally safe for students to walk around the campus. (CYUT Visit-A-1:50)*

*Our basic idea is probably to follow the hilly terrain, ..., so the buildings are generally in the north-south direction, then we leave the center empty, ..., we initially wanted to have it serve as an interior connection, like a corridor. However, the final product is not exactly like what we had originally planned... (STU Visit-B-2:57)*

*4.6. The Integration and Ecologicalization of the Campus Landscape and Open Space*

Concentrated and high-level design are being adopted by both schools for spatial layout under the situation given the limitation of practical usable area of school grounds and the reduction in floor space of the school buildings owing to cramped coverage and hillside terrain in the aspect of spatial scale according to case comparison. But for Shu-Te Institute of Technology, a large-sized banding green landscape zone is sited in the center of the campus for reason of soil horizon security concerns, which contrarily soothes "the sense of oppression" brought to campus users, either via inner feeling or on visual effects and sustains the availability of an adequate, pleasant scale space. As to the shaping of open space, restricted by the lay of the land, the campuses of both schools become isolated, accompanied inevitably by the lack of spatial liaison and fusion with town and urban areas. Furthermore, the schools fail to put up sports venue and facilities, which can most likely be open to and shared with city dwellers, at the site with strong communications with the outside world. Instead, the allocation takes place at the end of the campus, procuring weak accessibility and convenience that the authentic effectiveness of campus space's openness to the outside world is confined. Mentioned in the interview:

*After all, this is a private school which can't be compared to those national schools, ..., so it is very important to make good use of space... (CYUT Visit-A-1:35)*

*The main open space of the school is probably the square are in the center of the five older buildings... Because the number of primary school students in the school has changed a lot in the past few years, the spatial density has become very large and the outdoor space has thus become quite important... (CYUT Visit-A-1:38)*

*Because it is on the hillside, in fact, there is still a distance from the surrounding communities. The connection with the urban area may be weaker..., there is a check point at the entrance where you just entered. It is mainly used to control the entry and exit of vehicles. Community residents can also come to the campus for walks or exercises... (CYUT Visit-A-1: 62)*

*... The original idea was to concentrate the buildings as much as possible, make the buildings higher but reserve more land for green space or landscapes. Therefore, I placed all the buildings on the sides of the base surrounded by a relatively large green area. Since our land is on a hillside, the overall openness and landscape view will be very good. For a campus, this outdoor space is very important. It allows students the space for activities or socialization... Of course, people from outside of the campus can also come in,... (STU Visit-B-2: 64)*

## 5. Conclusions

For a university, the primary stage of school establishment is a very special phase of campus development. The main purpose is to make firm the framework of university campus with the capability of continuous growth, laying the groundwork for favorable development of the university. This study summarizes the six characteristics of campus planning development in the early stages of foundation of private university of science and technology in Taiwan.

However, according to research findings, the primary goal, which the private schools set during the early stages of foundation, is to expand actively the scale of school affairs under operating cost consideration. Not only has the number of new-founded departments, teachers and students been increasing year by year, the size of the school has been enlarged rapidly, which reaches up to 12,000 to 15,000 students, within 10 years after the establishment of the two cases. Therefore, the scale of the school outstrips the initial objective of middle and long-term school development in less than ten years after registration and institution. What is more, since private schools tend to run its business in line with the pragmatic enterprise management spirit that they devote particular care to fulfill the economic benefits of everything and to attain the maximum economic efficiency concerning the aspect of land utilization, the development strength of the land is prone to "high density development type". In virtue of the limitation of reserved space for development, the space of the campus soon become saturated that schools are forced to search for other grounds for augmentation during the rapid growth of school affairs. In other words, due to the infinite demand of school development and the limited resources of campus space, once the school development speed is too fast, it often causes the overload of campus land carrying capacity and school space tolerance; furthermore, resulting in the order gradually collapsed and the deterioration of the quality of the campus environment. It is thus easy to recognize the myth of private schools' pursuing proactively school growth, leading bit by bit to the emergence of the disequilibrium between "quality" and "quantity" of high TVE. Even more private schools should rethink and focus on seriousness.

And lastly, though the results of this study could merely indicate the performing progress and manifestation presented by the planning development of Taiwan's private technical and vocational institutes in the pioneering days of foundation, the skeleton and outline of research outcomes can still serve as the basis for the comprehensive study on the campus of university of technology in the future. On the other hand, the 2015 UN General Assembly continued the previous "Millennium Development Goals (MDGs)" and drafted the "Sustainable Development Goals (SDGs)" based on active practice of equality and human rights and takes into account "economic growth", "social progress" and "environmental protection". Among them, it is not hard to find that "people" is the core of sustainable development and "education" is the key to the success of each sustainable development goal. According to a survey organized by the Higher Education Sustainable Development Initiative (HESI), collaborating with universities is the most effective way to promote sustainable development goals (SDGs). Universities play an important role in cultivating talents, disseminating knowledge and educating the society. It is not only the best practice platform for SDGs but also the content of the SDGs project. Therefore, for the university campus as an educational carrier, it is an important experimental field for practicing the concept of sustainable development. Promoting sustainable development on the university campus can be an effective way to build a sustainable future for our next generation.

**Author Contributions:** C.-J.S. contributed to the conceptual design of the study, data collection, drafting the article, and final approval. S.-C.C. contributed to the conceptual design of the study, supervision of the progress, and final approval.

**Funding:** This research was funded by Ministry of Science and Technology, grant number MOST 103-2410-H-224-018.

**Acknowledgments:** The authors would like to thank the editor and anonymous reviewers for their constructive comments and valuable suggestions to improve the paper. Also, sincere thanks for all interviewees for sharing their valuable experiences and contributing to this research. The support of the Ministry of Science and Technology (project MOST 103-2410-H-224-018) is gratefully acknowledged.

**Conflicts of Interest:** The authors declare no conflict of interest.

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
