# Peer review of "The Comparison of Campus Planning Development at the Initial Stage of School Establishment: A Study of the Two Newly Instituted Private Universities of Science and Technology in Taiwan"

_sustainability, doi:10.3390/su11061525_

Round 1
Reviewer 1 Report
The article focuses, from two case studies focused on technical education, within the spatial evolution of the campus in its initial stage. The topic is of interest to detect some conditioning aspects of the final evolution of the campuses. The two cases of study show very different aspects, but, in a global way, they allow to identify planning guidelines that show similar strategies.
The spatial pressure suffered by the rapid institutional growth leads to a progressive increase in the rate of land use. It is clearly observed that the starting conditions are altered by a higher density and therefore a greater limitation in the free spaces.
The six aspects raised in the study of the cases are perfectly valid and solid. The proposed analysis allows detecting the main characteristics for later comparison. Thus, the method used is strong and can be extrapolated to other areas of research.
I provide some comments on the development of the article:
The list of keywords I think should be more precise, for example, better to use "private university" than "private university of technology".
Regarding the introduction, the increase in universities is explained, but no direct sources are provided to corroborate the data. If the data is from own analysis, explain how it was obtained. When was it regulated by the Ministry of Education? You must indicate the source or the law in the list of references. When the article talks about 79% of the students are in technical universities, the source is not indicated. An acronym for "colleges of technology / universities of science & technology" would help with reading.
In 2.1 it would be interesting that the authors have these two publications as reference:
Lin, C.-Y., Hsieh, Y.-H., Chen, C.-H. (2014). Taiwanese college instructors’ evaluation of their schools: The differences among school attributes and instructor ranks. Journal of Research in Education Sciences 59(3), pp. 29-58.
Lin, M.-H., Hu, J., Tseng, M.-L., Chiu, A.S.F., Lin, C. (2016). Sustainable development in technological and vocational higher education: Balanced scorecard measures with uncertainty. Journal of Cleaner Production 120, pp. 1-12
On line 173, a possible text error appears: lo-cation.
In 2.3. Some inspiring articles would be appropriate to take into consideration:
Yen-Yi Li; Hsin-Hao Chen; Wen-Cheng Shao; Shuh-Ren Jing; Fong-Yu Chiu; Huey-Jen Jenny Su (2017). Practices of innovative technology and education for sustainability in Taiwan sustainable campus program. Pacific Neighborhood Consortium Annual Conference and Joint Meetings (PNC). Doi: 10.23919/PNC.2017.8203517
Yi Fan, Kenichi Tanoue (2018). A study on the interaction between schools of New Campus Movement and local communities in Taiwan. JournalAIJ Journal of Technology and Design, 24(56): 301-306. Doi: 10.3130/aijt.24.301
The literal reference 10 should indicate the page.
In section 2.4. there are several issues that are left aside. In recent years, the vision of a campus design connected with the environment leads us to models where sustainability is a determining factor:
Alshuwaikhat, H.M., Abubakar, I. (2008). An integrated approach to achieving campus sustainability: assessment of the current campus environmental management practices. Journal of cleaner production. Elsevier.
Sharp, L. (2009). Higher education: the quest for the sustainable campus. Taylor & Francis.
On the other hand, the participation of different stakeholders in the planning process determines the design of the campus and gives a new quality to a holistic vision of the campus (physical and spiritual):
Ribalaygua Batalla, C., Garcia Sanchez, F. (2016). Creating a Sustainable Learning District by Integrating Different Stakeholders’ Needs. Methodology and Results from the University of Cantabria Campus Master Plan. In alter Leal Filho, Luciana Brandli. (eds). Engaging Stakeholders in Education for Sustainable Development at University Level. World Sustainable Series. Springer, pp. 3-20.
J. Davis (2010). Participatory design for sustainable campus living. CHI'10 Extended Abstracts on Human Factors see www.dl.acm.org.
Regarding the case studies:
Why are figures 3, 4 and 5 not facing north? Is it perhaps due to the texts located inside the figures? It seems necessary to make an effort to properly graph the campus, with its northern orientation. Your own figures are more interesting than those obtained in other documents.
In figure 3 a legend must be included that explains the colors used (functions and attributes indicated in the main text)
For a correct understanding of the information, the color code and arrows used in figures 3, 4 and 5 must be repeated in figures 8 (functions), 9 and 10.
Why are they not oriented correctly? Why the picture is oriented to south?
The project (plan) represented in these images do not correspond exactly to the current situation (see Google Maps). Why? Is it an initial project? Why has it changed since its initial phase? An analysis must also be made of the current situation.
In section 4.5 there is no mention of the extensive parking lots, especially motorcycles. This aspect greatly conditions the spatial morphology of the campus.
Final comments:
Images 6 and 11 must be modified. They do not have enough quality. It would be convenient for authors to have their own images.
I believe that a broader review of the bibliography on campus design and TVE is necessary.
Author Response
Response to Reviewer 1 Comments
Dear Sir:
Thank you for inviting us to submit a revised draft of our manuscript entitled, "The Comparison of Campus Planning Development at the Initial Stage of School Establishment – A study of the two newly instituted private universities of science and technology in Taiwan" to Sustainability. We also appreciate the time and effort you and each of the reviewers have dedicated to providing insightful feedback on ways to strengthen our paper. Thus, it is with great pleasure that we resubmit our article for further consideration. We have incorporated changes that reflect the detailed suggestions you have graciously provided. We also hope that our edits and the responses we provide below satisfactorily address all the issues and concerns you and the reviewers have noted.
To facilitate your review of our revisions, the following is a point-by-point response to the questions and comments delivered in your letter dated 07 February 2019.
Reviewer 1 Comments
Point 1: The list of keywords I think should be more precise, for example, better to use "private university" than "private university of technology".
Response 1: Thank you for your suggestion. We have replaced the term "private university of technology" throughout the paper with "private university" to use more precise terms.
Point 2: Regarding the introduction, the increase in universities is explained, but no direct sources are provided to corroborate the data. If the data is from own analysis, explain how it was obtained. When was it regulated by the Ministry of Education? You must indicate the source or the law in the list of references. When the article talks about 79% of the students are in technical universities, the source is not indicated.
Response 2: Thank you for your suggestion. We have provided additional explanation in Section 1.1. Additionally, we have also added “Table 1. Statistics of Universities of Technology & Colleges of Technology”and “Table 2. Statistics of Vocational Universities Enrolment over the Years,”
Point 3: An acronym for "colleges of technology / universities of science & technology" would help with reading..
Response 3: Thank you for your suggestion. We have corrected it accordingly.
Point 4: In 2.1 it would be interesting that the authors have these two publications as reference:
Lin, C.-Y., Hsieh, Y.-H., Chen, C.-H. (2014). Taiwanese college instructors’ evaluation of their schools: The differences among school attributes and instructor ranks. Journal of Research in Education Sciences 59(3), pp. 29-58.
Lin, M.-H., Hu, J., Tseng, M.-L., Chiu, A.S.F., Lin, C. (2016). Sustainable development in technological and vocational higher education: Balanced scorecard measures with uncertainty. Journal of Cleaner Production 120, pp. 1-12
Response 4: Thank you for providing these insights. We have incorporated this suggestion throughout our paper in section 2.1.
Point 5: On line 173, a possible text error appears: lo-cation.
Response 5: Thank you for your suggestion. We have corrected it accordingly.
Point 6: In 2.3. Some inspiring articles would be appropriate to take into consideration:
Yen-Yi Li; Hsin-Hao Chen; Wen-Cheng Shao; Shuh-Ren Jing; Fong-Yu Chiu; Huey-Jen Jenny Su (2017). Practices of innovative technology and education for sustainability in Taiwan sustainable campus program. Pacific Neighborhood Consortium Annual Conference and Joint Meetings (PNC). Doi: 10.23919/PNC.2017.8203517
Yi Fan, Kenichi Tanoue (2018). A study on the interaction between schools of New Campus Movement and local communities in Taiwan. JournalAIJ Journal of Technology and Design, 24(56): 301-306. Doi: 10.3130/aijt.24.301
Response 6: This is an interesting perspective. We agree with you and have incorporated this suggestion throughout our paper in section 2.3.
Point 7: The literal reference 10 should indicate the page.
Response 7: Thank you for your suggestion. We have corrected it accordingly.
Point 8: In section 2.4. there are several issues that are left aside. In recent years, the vision of a campus design connected with the environment leads us to models where sustainability is a determining factor:
Alshuwaikhat, H.M., Abubakar, I. (2008). An integrated approach to achieving campus sustainability: assessment of the current campus environmental management practices. Journal of cleaner production. Elsevier.
Sharp, L. (2009). Higher education: the quest for the sustainable campus. Taylor & Francis.
On the other hand, the participation of different stakeholders in the planning process determines the design of the campus and gives a new quality to a holistic vision of the campus (physical and spiritual):
Ribalaygua Batalla, C., Garcia Sanchez, F. (2016). Creating a Sustainable Learning District by Integrating Different Stakeholders’ Needs. Methodology and Results from the University of Cantabria Campus Master Plan. In alter Leal Filho, Luciana Brandli. (eds). Engaging Stakeholders in Education for Sustainable Development at University Level. World Sustainable Series. Springer, pp. 3-20.
J. Davis (2010). Participatory design for sustainable campus living. CHI'10 Extended Abstracts on Human Factors see www.dl.acm.org.
Response 8: Thank you for providing these insights. We agree with you and have incorporated this suggestion throughout our paper in section 2.4.
Point 9: Why are figures 3, 4 and 5 not facing north? Is it perhaps due to the texts located inside the figures? It seems necessary to make an effort to properly graph the campus, with its northern orientation. Your own figures are more interesting than those obtained in other documents.
Response 9: Thank you for your suggestion. We have corrected it accordingly. We hope these revisions provide a more clear understanding of the information.
Point 10: In figure 3 a legend must be included that explains the colors used (functions and attributes indicated in the main text)
Response 10: Thank you for your suggestion. We have corrected it accordingly. We hope these revisions provide a more clear understanding of the information.
Point 11: For a correct understanding of the information, the color code and arrows used in figures 3, 4 and 5 must be repeated in figures 8 (functions), 9 and 10.
Response 11: Thank you for your suggestion. We have corrected it accordingly. We hope these revisions provide a more clear understanding of the information.
Point 12: Why are they not oriented correctly? Why the picture is oriented to south?
Response 12: We agree with you. We have corrected it accordingly. We hope these revisions provide a more clear understanding of the information.
Point 13: The project (plan) represented in these images do not correspond exactly to the current situation (see Google Maps). Why? Is it an initial project? Why has it changed since its initial phase? An analysis must also be made of the current situation.
Response 13: You have raised an important question. The initial campus planning is to draft a blueprint for the overall spatial architecture of a university which may be partially modified as the school grows and brings in different architects to design new buildings. However, there will be no major changes in the overall planning of the campus. In this case, the campus plan you saw is the original version of the campus plan which is different from you would see on Google Map. Therefore, in order to make the message more accurate, we added the current campus map to illustrate. We hope that this will answer your questions.
Point 14: In section 4.5 there is no mention of the extensive parking lots, especially motorcycles. This aspect greatly conditions the spatial morphology of the campus.
Response 14: Thank you for your suggestion. We agree with you and have incorporated this suggestion throughout our paper in section 4.5.
Point 15: Images 6 and 11 must be modified. They do not have enough quality. It would be convenient for authors to have their own images.
Response 15: Thank you for your suggestion. We agree with you. The Academic Editor also mentioned that some photos in the article are not necessary. Therefore, we removed Images 6 and 11 and replaced Images 1 and 2 with pictures taken from Google Earth to provide more detailed information about the two colleges.
Point 16: I believe that a broader review of the bibliography on campus design and TVE is necessary.
Response 16: Thank you for your suggestion. We have provided a more in-depth discussion on the bibliography on campus design and TVE in Sections 2.1, 2.3 and 2.4.
CONCLUDING REMARKS:
Thank you once again for giving us the opportunity to strengthen our manuscript with your valuable suggestions. We have done our best to incorporate your feedback and sincerely hope that these revisions will persuade you to accept our submission.
Sincerely,
Chuan-Jen Sun

Reviewer 2 Report
Dear Editor, Dear Authors
This is a very interesting, science pertinent and sustainability timely contribution;
Some remarks to be addressed:
i) There is a need to reinforce the theoretical referential /state of the art about campus sustainability and sustainability barriers (see for instance the papers from Aleixo et al., in the Journal of Cleaner Production and Farinha et al., and Aleixo et al., in the International Journal of Sustainability in Higher Education);
ii) Bibliography is limited, mostly grey literature;
iii) There is no data/results about the conducted interviews;
iv) Maybe some discussion /Future perspectives in the context of the SDGs will make this contribution even more interesting.
Thanks for the opportunity
Author Response
Response to Reviewer 2 Comments
Dear Sir:
Thank you for inviting us to submit a revised draft of our manuscript entitled, "The Comparison of Campus Planning Development at the Initial Stage of School Establishment – A study of the two newly instituted private universities of science and technology in Taiwan" to Sustainability. We also appreciate the time and effort you and each of the reviewers have dedicated to providing insightful feedback on ways to strengthen our paper. Thus, it is with great pleasure that we resubmit our article for further consideration. We have incorporated changes that reflect the detailed suggestions you have graciously provided. We also hope that our edits and the responses we provide below satisfactorily address all the issues and concerns you and the reviewers have noted.
To facilitate your review of our revisions, the following is a point-by-point response to the questions and comments delivered in your letter dated 07 February 2019.
Reviewer 2 Comments
Point 1: There is a need to reinforce the theoretical referential /state of the art about campus sustainability and sustainability barriers (see for instance the papers from Aleixo et al., in the Journal of Cleaner Production and Farinha et al., and Aleixo et al., in the International Journal of Sustainability in Higher Education)
Response 1: Thank you for providing these insights. We have incorporated this suggestion throughout our paper in section 2.
Point 2: Bibliography is limited, mostly grey literature
Response 2: Thank you for your suggestion. We have provided a more in-depth discussion on the bibliography on campus design and TVE in Sections 2.1, 2.3 and 2.4. We hope these revisions provide a more thorough discussion.
Point 3: There is no data/results about the conducted interviews
Response 3: Thank you for your suggestion. We have provided additional data of the interviewees in Section 4. We hope these revisions provide a more thorough discussion. We hope these revisions provide a more thorough discussion.
Point 4: Maybe some discussion /Future perspectives in the context of the SDGs will make this contribution even more interesting.
Response 4: Thank you for providing these insights. We agree with you and added additional discussion at the end of Section 5. We hope these revisions provide a more thorough discussion.
CONCLUDING REMARKS:
Thank you once again for giving us the opportunity to strengthen our manuscript with your valuable suggestions. We have done our best to incorporate your feedback and sincerely hope that these revisions will persuade you to accept our submission.
Sincerely,
Chuan-Jen Sun
